# A Novel Missense Mutation in *ERCC8* Co-Segregates with Cerebellar Ataxia in a Consanguineous Pakistani Family

**DOI:** 10.3390/cells11193090

**Published:** 2022-09-30

**Authors:** Zeeshan Gauhar, Leon Tejwani, Uzma Abdullah, Sadia Saeed, Shagufta Shafique, Mazhar Badshah, Jungmin Choi, Weilai Dong, Carol Nelson-Williams, Richard P. Lifton, Janghoo Lim, Ghazala K. Raja

**Affiliations:** 1University Institute of Biochemistry and Biotechnology (UIBB), Pir Mehr Ali Shah Arid Agriculture University Rawalpindi (PMAS-AAUR), Rawalpindi 46300, Pakistan; 2Interdepartmental Neuroscience Program, Yale School of Medicine, New Haven, CT 06510, USA; 3Department of Neuroscience, Yale School of Medicine, New Haven, CT 06510, USA; 4Department of Health Informatics, Women University, Swabi 23430, Pakistan; 5Department of Neurology, Pakistan Institute of Medical Sciences (PIMS), Islamabad 04485, Pakistan; 6Department of Genetics, Yale School of Medicine, New Haven, CT 06510, USA; 7Department of Biomedical Sciences, Korea University College of Medicine, Seoul 02841, Korea; 8Laboratory of Human Genetics and Genomics, Rockefeller University, New York, NY 10065, USA; 9Program in Cellular Neuroscience, Neurodegeneration and Repair, Yale School of Medicine, New Haven, CT 06510, USA

**Keywords:** autosomal-recessive cerebellar ataxias, ARCA, cerebellum, ERCC8

## Abstract

Autosomal-recessive cerebellar ataxias (ARCAs) are heterogeneous rare disorders mainly affecting the cerebellum and manifest as movement disorders in children and young adults. To date, ARCA causing mutations have been identified in nearly 100 genes; however, they account for less than 50% of all cases. We studied a multiplex, consanguineous Pakistani family presenting with a slowly progressive gait ataxia, body imbalance, and dysarthria. Cerebellar atrophy was identified by magnetic resonance imaging of brain. Using whole exome sequencing, a novel homozygous missense mutation *ERCC8*:c.176T>C (p.M59T) was identified that co-segregated with the disease. Previous studies have identified homozygous mutations in *ERCC8* as causal for Cockayne Syndrome type A (CSA), a UV light-sensitive syndrome, and several ARCAs. *ERCC8* plays critical roles in the nucleotide excision repair complex. The p.M59T, a substitution mutation, is located in a highly conserved WD1 beta-transducin repeat motif. In silico modeling showed that the structure of this protein is significantly affected by the p.M59T mutation, likely impairing complex formation and protein-protein interactions. In cultured cells, the p.M59T mutation significantly lowered protein stability compared to wildtype ERCC8 protein. These findings expand the role of *ERCC8* mutations in ARCAs and indicate that ERCC8-related mutations should be considered in the differential diagnosis of ARCAs.

## 1. Introduction

Autosomal-recessive cerebellar ataxias (ARCAs) comprise a heterogeneous group of rare neurological disorders associated with atrophy of the cerebellum and its associated tracts [1,2]. Disease symptoms can manifest during childhood or in adulthood before the age of 40, including cerebellum-associated features of uncoordinated movements such as imbalanced gait, dystonia, dysarthria, and chorea etc. [3]. Clinical presentation can also involve other neurological (retina, cerebral cortex, basal ganglia, corticospinal tracts, and peripheral nerves, etc.) and non-neurological (heart, pancreas, and muscle, etc.) systems [2].

The genetic etiology of ARCAs is diverse but partially known. Recent advances in high throughput sequencing methods, especially whole exome sequencing (WES), have led to a rapid increase in the discovery of ARCA genes, with over 92 genes identified for disorders in which ataxia is present or a predominant feature [4,5]. This accounts for about 50% cases of ataxias with a suspected diagnosis of ARCAs [6]. Although ARCAs caused by different genes show clinical heterogeneity, the disease-causing genes share some common molecular mechanisms and cellular pathways such as impaired DNA repair, disturbed mitochondrial metabolism, or defective phospholipid remodeling [6]. For a better understanding of cerebellar physiology, ARCAs can also be grouped based on deficient cellular and metabolic pathways. Defects in the DNA repair mechanism involving double-strand break-repair pathway or single-strand break-repair complexes are a predominant pathological mechanism of several ARCAs. DNA damage can also lead to cerebellar degeneration, through mechanisms that are not yet fully understood [7].

One important DNA repair pathway, nucleotide excision repair, is mediated by the protein product of the *ERCC8*/*CSA* gene [6] located on chromosome 5q12.1. *ERCC8* is comprised of 12 exons, which encode a 396-amino acid ERCC8/CSA protein containing seven highly conserved repeated WD40 motifs (short~40 amino acids residue often terminating in a Trp-Asp (*W*-*D*) dipeptide) [8,9]. The dysfunction of ERCC8 has previously been linked with Cockayne syndrome (CS) [10] and a UV-light sensitive syndrome (UVSS) [11]. CS is characterized by growth failure, progressive neurologic dysfunction, microcephaly, and intellectual disability along with other defects such as cutaneous photosensitivity, kyphosis, ankylosis, and optic atrophy [12]. This broad phenotypic spectrum reveals the importance of the normal function of ERCC8 in many organs and cell types, which is reflected in its involment in different disorders. Nonetheless, there is variable expressivity of the constellation of phenotypes resulting from *ERCC8* mutations, and phenotype-genotype correlations remain elusive. Missense mutations in *ERCC8* have been reported to cause milder phenotypes than protein truncating mutations [10], and ataxia can be part of a broad syndromic constellation of phenotypes [13,14] or can appear as an isolated clinical feature [15]. A total of 38 mutations have so far been identified in *ERCC8*, which are largely composed of missense mutations and deletions [14]. In this study we identified a novel homozygous missense mutation c.176T>C in the *ERCC8* gene that introduces a previously unreported M59T mutation in the encoded protein in a consanguineous ARCA Pakistani family with three affected male siblings.

## 2. Material and Methods

### 2.1. Participants and Procedure

The ethical approval for the study was obtained (Ethics Committee for Use of Human Subjects, Pir Mehr Ali Shah Arid Agriculture University Rawalpindi, Pakistan PMAS-AAUR/IEC-14/2016). A consanguineous Pakistani family (Figure 1), comprising three affected and eleven unaffected individuals, was recruited from the Department of Neurology and Neurophysiology, Pakistan Institute of Medical Sciences (PIMS), Islamabad, Pakistan. The affected siblings were diagnosed based on clinical symptoms and brain magnetic resonance imaging (MRI). A written informed consent was obtained from all the participating individuals or the parents (in case of minors) along with a complete medical history (Table 1). Venous blood samples were drawn from patients and unaffected family members, and collected in EDTA-coated vacutainers. Genomic DNA was extracted by using whole blood genomic GeneJetDNA purification mini kit (ThermoScientific, Cat# GeneJet K0782; Waltham, MA, USA).

### 2.2. Genetic Analysis

In order to identify the disease-causing variants, whole exome sequencing (WES) was performed at the Yale Center for Genome Analysis (YCGA) using DNA from two affected siblings IV:2 and IV:3. The sample DNA was fragmented, linkers ligated followed by agarose gel electrophoresis for fractionation. The extracted DNA was amplified using Polymerase Chain Reaction (PCR) and hybridized to the capture arrays. After removing the non-hybridized DNA by washing the bound DNA was eluted and ligation mediated PCR was performed. The PCR products were purified and processed for DNA sequencing on the Illumina platform (San Diego, CA, USA). Image analysis and base calling were done after sequencing the captured libraries on the Illumina genome analyzer. Burrows-Wheeler Aligner software (BWA Software, v0.7.11, San Diego, CA, USA) was used to map the sequence reads with reference genome (hg19) and SAMtools (v0.1.19, Harvard University, Cambridge, MA, USA) was used for the processing of resulting sequence data. Insertions/deletions (indels) were subsequently filtered against reference genome. Public databases used as a filter were 1000 Genomes, db-SNP and Exome Aggregation Consortium’s ExAC Browser. Given the presumptive evidence of recessive transmission of the trait in the pedigree, genes with biallelic variants having minor allele frequency (MAF) < 0.01 in public databases of genomic variants (gnomAD, 1000 Genomes) were identified. The variants were further annotated according to the predicted pathogenicity scores of multiplein silicotools such as PolyPhen2 (http://genetics.bwh.harvard.edu/pph2 (accessed on 18 October 2016)), MetaSVM, SIFT (http://sift.jcvi.org (accessed on 18 October 2016)) and Combined Annotation Dependent Depletion (CADD; https://cadd.gs.washington.edu/ (accessed on 18 October 2016)) score.

To validate identified *ERCC8* mutations and to check its segregation in the remaining family members, PCR amplification followed by Sanger sequencing was performed (Keck DNA sequencing facility, Yale University, New Haven, CT, USA). The primers, forward 5′-CCTGCCAATGGAACACACTG-3′ and reverse 5′-CCGTCTTGTTTCTGTCACCC-3′, were designed using primer-BLAST. PCR amplification was performed using 11 μL ddH_2_O, 2 μL 10× PCR buffer (Invitrogen, Waltham, MA, USA), 2.5 μL of 0.2 mM dNTPs, 1 μL of 0.5 μM primers, 0.5 μL Taq DNA polymerase (Invitrogen), 0.5 μL DMSO and 2 μL genomic DNA. The thermal profile of amplification reaction was as follows: Initial denaturation at 95 °C for 5 min followed by 30 cycles of 95 °C for 30 s, 67 °C for 30 s, and 72 °C for 45 s, with a final elongation of 72 °C for 5 min in a Bio-Rad DNA Thermal Cycler (Hercules, CA, USA). The PCR products were confirmed on a 1.5% agarose gel and purified by QIAGEN PCR purification kit (Cat#28104S; Hilden, Germany). Sanger sequencing was performed and electropherograms were visualized using ChromasPro 1.7.6 software (Technelysium, South Brisbane, Australia).

### 2.3. In Silico Modelling of Mutant Protein

To determine the predicted consequences of identified mutations on the structure of ERCC8 protein, crystal structures of human ERCC8 protein (PDB entry: 6FCV) were retrieved through protein data bank (PDB) followed by energy minimization using amber force field embedded in UCSF Chimera 1.10.0 [16]. In order to explore the effect of mutated ERCC8, 3D structure of ERCC8^M59T^ was predicted via MODELLER 9.13 tool [17] using wildtype (WT) ERCC8 (ERCC8^WT^) as a reference template. MolProbitytool (Version 4.5.1, Department of Biochemistry, Duke University, Durham, NC, USA) (http://molprobity.biochem.duke.edu/ (accessed on 30 June 2021)) [18] was utilized to validate the 3D models while geometry optimization was carried out using YASARA (http://www.yasara.org/minimizationserver.htm/ (accessed on 30 June 2021)) [19].

### 2.4. Biochemical Analysis of Mutant ERCC8

Overexpression vectors containing WT *ERCC8* transcript variant 1 cDNA were purchased (Origene, Rockville, MD, USA; Cat# RC214392) and site-directed mutagenesis was performed (Agilent QuikChange II XL; Cat# 200521; Agilent Technologies, Santa Clara, CA, USA) to generate mutant *ERCC8* expression vectors using following primers: forward 5′-TCTGAACCACCTGATAACGTGTATCTCCCTTCAACAG-3′ and reverse 5′-CTGTTGAAGGGAGATACACGTTATCAGGTGGTTCAGA-3′. HeLa cells were transfected with WT or mutant *ERCC8*. Forty-eight hours later, transfected cells were treated with 100 μg/mL cycloheximide (CHX) for two to eight hours to inhibit new protein translation. Cells were collected in triplicates every two hours during the time course and subsequently processed for Western blotting as previously described [20] with slight modifications. Briefly, cells were pelleted and lysed in lysis buffer (50 mM Tris [pH 7.5], 150 mM NaCl, 0.1% SDS, 0.5% Triton X-100, 0.5% NP-40 and Roche complete protease inhibitor cocktail), rotated at 4 °C for 30 min, and then centrifuged for 15 min at 13,000 rpm at 4 °C. 20 μg total protein for each sample was loaded onto an SDS-PAGE gel, transferred onto a nitrocellulose membrane, washed three times in TBST, blocked by incubation in 5% skimmed milk in TBST, and incubated with primary antibodies (mouse anti-Vinculin, Sigma (St. Louis, MO, USA), V9264, 1:10,000; rabbit anti-ERCC8, Abcam (Cambridge, UK), ab137033, 1:1000) diluted in 5% skimmed milk in TBST overnight. The following day, membranes were washed three times in TBST, incubated with HRP-conjugated secondary antibodies for one hour, and detected using ECL reagents.

## 3. Results

### 3.1. Clinical Presentations

In the family studied here, three of four siblings from a first-cousin union presented with progressive ataxia (Figure 1A). All four siblings were born following full-term, uneventful pregnancies. The presence of consanguinity and the absence of ataxia in both parents, as well in the four offspring of one of the affected siblings, and the absence of ataxia among other members of the kindred suggest a strong likelihood of recessive transmission of the phenotype in this family. The three affected members achieved developmental milestones normally and clinical history was unremarkable except for blister formation in members IV:1, IV:2 and IV:3, resulting from prolonged sunlight exposure. At later ages (10, 12, and 15 years, respectively), the affected individuals developed involuntary movements of hands and difficulty in handling objects. Siblings also presented with slowly progressing neurological symptoms such as dysarthria and gait ataxia, testing positive upon neurological examination using the finger-nose and heel-shin tests. During examination for the present study (Table 1), the ages of all three affected subjects were as follows; IV:1 30 years, IV:2 37 years, and IV:3 43 years. All affected members showed impaired coordination of legs and arms, difficulty in holding objects, dysphagia, and diplopia. However, reflexes were intact in subject IV:1, while depressed in IV:2 and IV:3, with normal cognition in all patients. Importantly, characteristic features of CS-like microcephaly, growth failure, kyphosis, ankylosis, and optic atrophy were absent in all patients.

### 3.2. Magnetic Resonance Imaging (MRI)

Brain MRIs of all three affected and one healthy control (Figure 1B) were performed on a 3 Tesla MRI machine at Radiology Department, Military Hospital (MH), Rawalpindi, Pakistan. The MRI sequences of the affected individual IV:3 showed prominent cerebellar atrophy/degeneration with prominent intra and extra cerebrospinal fluid (CSF) spaces as shown on axial section of T2W1. The MRI sequences of the affected subject IV:2 also showed prominent cerebellar atrophy/degeneration but with mildly dilated ventricles and evidence of loss of cerebral volume, suggestive of cerebral atrophy on axial section of T2W1. For the third affected sibling IV:1 (the youngest patient), MRI showed very mild atrophy of cerebellar folia on T2W1, that is proposed to be age-related. The IV:1 member also had a blooming artifact on gradient echo sequence assumed to be the result of calcification.

### 3.3. Genetic Findings

The prioritization criteria for rare biallelic variants generated a list of four genes as shown in Table 2 with the same biallelic variants in both sequenced siblings, all of which were homozygous. One of these genes, *ERCC8*, had previously been implicated in recessive ataxia, and had a homozygous missense variant that has not previously been implicated in disease: *ERCC8* (RefSeq NM_000082: c.176T>C (p.M59T). Sanger sequencing revealed complete co-segregation of homozygosity of this variant with the disease phenotype in this family. All three affected siblings were homozygous for the variant and all 5 of the unaffected individuals at risk of inheriting the recessive genotype were heterozygotes. The odds of this co-segregation occurring by chance alone were 1047:1 (Lod score 3.02), meeting traditional thresholds for significant linkage. Other healthy family members were either heterozygous carriers or homozygous for the WT allele (Figure 2A). The c.176T>C (p.M59T) mutation is located in the third exon of the full-length isoform and substitutes for a conserved amino-acid localized in the WD1 domain of ERCC8 protein. Comparative amino acid alignment of the protein across different species revealed that this amino acid has been highly conserved during evolution (Figure 2B). However, no ortholog is present in *Drosophila*, while orthologs of *Caenorhabditis elegans* with 18% identity and *Saccharomyces cerevisiae* with 11% identity where the M59 is not conserved in both of these.

### 3.4. In Silico Analysis

In order to obtain the complete 3D structure of ERCC8^M59T^, a comparative modeling technique was employed using ERCC8^WT^ as template. Ramachandran plot designated the presence of more than 95.69% residues of ERCC8^M59T^-modeled structure in the sterically allowed region, while 0.76% residues were considered as outliers. Only a few poor rotamers were observed. Parameters including peptide bond planarity, non-bonded interactions, Cα-tetrahedral distortion, main chain H-bond energy, and the overall G-factor for the modeled structure were within the favorable range. ERCC8^M59T^ structure was minimized through Yasara. To explore the effect of ERCC8^M59T^ mutation, the predicted ERCC8^M59T^ structure was superimposed with the template (ERCC8^WT^) by UCSF Chimera. An RMSD value of 0.45Å indicated a significant conformational changes upon mutation (Figure 2C). In contrast to ERCC8^WT^, β-strand extended three residues (Gln326-Lys331) upon mutation. Particularly, upon mutation the His180-Leu182 loop region was inferred to assume a β-strand conformation. Together, thisin silico modeling predicted that the mutation presumably alters the structure of the WD1 domain (Figure 2C).

### 3.5. M59 Mutation Destablizes ERCC8 Protein

Becausein silico modelling of variants predicted that the identified c.176T>C (p.M59T) mutation alters the protein structure (Figure 2C), we next wanted to explore whether this mutation affects ERCC8/CSA protein stability. To do so, we first generated cDNA clones to overexpress WT or mutant ERCC8/CSA (Figure 3A) and introduced these vectors into HeLa cells (Figure 3B), which resulted in robust overexpression above endogenous ERCC8 levels (data not shown). Cycloheximide (CHX) chase experiments over eight hours revealed enhanced rate of protein degradation of mutant ERCC8/CSA compared to WT counterpart. These data demonstrate that the identified mutation (p.M59T) in ERCC8/CSA results in loss of normal ERCC8/CSA function, at least in part due to reduced protein stability. 

## 4. Discussion

In this study, we identified a novel homozygous missense mutation *ERCC8*:c176T>C (p.M59T) segregating with cerebellar ataxia in a consanguineous Pakistani family. We used WES, which has previously identified novel genes involved in rare neurological disorders including ataxia, especially in the setting of consanguinity [21]. Mutations in *ERCC8* have been associated with defects in the transcription-coupled nucleotide excision repair (TC-NER) leading to CSA [22], characterized mainly by growth failure, microcephaly and developmental delay [23]. None of the three affected members of studied family showed classical symptoms of CS upon thorough examination by a neurologist and were excluded as having CS. In addition to CS, variants of *ERCC8* has also been linked to UV-sensitive syndrome (UVSS), which is restricted only to photosensitivity and pigmentation and absence of any neurological anomalies [24]. The mutant *ERCC8* gene has recently been reported as a cause of ARCA as a missense mutation p.Gly257Arg has been linked to unique cerebellar ataxia in a Chinese family [15]. All the three affected members of current study manifest classical symptoms of ARCA, including an early disease onset and characteristic, slowly progressive clinical features of cerebellar ataxia with mild skin photosensitivity. Brain MRIs of all subjects showed mild to moderate cerebellar atrophy in addition to cerebral involvement.

The *ERCC8*:c176T>C (p.M59T)variant identified in this Pakistani family is absent in gnomAD and to our knowledge, homozygous mutations altering M59 have not been reported in any of the public databases of human polymorphisms. Interestingly, this variant occurs in the initiation codon, and predicted to cause start-loss variant for two other *ERCC8* transcript isoforms (ENST00000426742.2 and ENST00000439176.1); however, these isoforms are not enriched in the brain at any stage of development (as shown in Appendix A, adopted from Genotype-Tissue Expression (GTEx) database); therefore loss of expression of these isoforms is unlikely to contribute to the clinical symptoms observed in this family. Hence, the pathology resulting from this mutation is likely due to by substitution mutation p.M59T, which affects the dominant, full-length *ERCC8* transcript. It is predicted to be pathogenic by multiple in silico tools. First, p.M59T is located in one of the WD repeats of CSA protein known to be involved in protein complex formation and other protein-protein interactions. The integrity of WD domains plays a key role in the biological functions of ERCC8/CSA protein as most of the reported mutations reside in these conserved domains [25]. ERCC8/CSA interacts with DDB1, Cullin 4A, and Roc1, and exhibits ubiquitin ligase activity in TC-NER [26]. The alteration in protein’s structure of ERCC8/CSA, as predicted by in silico analysis, can thereby assumed to affect its associations with other proteins [27]. For example, a missense mutation (Ala205Pro) reported in the fourth WD40 motif abolished interaction with DDB1 [27].

Recently, it has been reported that ubiquitination of RNAPII activates TC-NER and processing of DNA damage-stalled RNAPII, both of which provide a protective mechanism against progressive neurodegeneration. Furthermore, severely reduced ubiquitination has been observed in CSA-deficient cells [28]. In cultured HeLa cells, we showed that in comparison to ERCC8/CSA^WT^, the ERCC8/CSA^M59T^ mutant protein is less stable and has a faster turnover. Therefore, the reduced ubiquitination observed in CSA mutant cells can potentially be attributed to both lower protein levels due to protein instability, as well as a reduction of ubiquitin ligase activity resulting from the p.M59T mutation in the functional domain of the remaining protein. Although these experiments suggest that p.M59T mutation results in a loss of ERCC8 function, further biochemical studies with larger sample sizes that discriminate between endogenous WT and exogenous WT and mutant ERCC8 are required to conclusively determine the impact of this mutation on protein stability.

The p.M59T mutation identified here resides in the WD1 repeat motif where no missense mutation has been reported previously. However, additional functional investigations are required to elucidate the precise effect of c.176T>C variant on *ERCC8* gene function. To the best of our knowledge, this is the first report of a Pakistani family linking an *ERCC8* mutation to ARCAs, broadening the spectrum of clinical phenotypes of the pathogenic variants associated with ARCAs. It is also the first report of the clinical consequences of c.176T>C mutation in the WD1 repeat and highlights the need for further functional studies.

## Figures and Tables

**Figure 1 cells-11-03090-f001:**
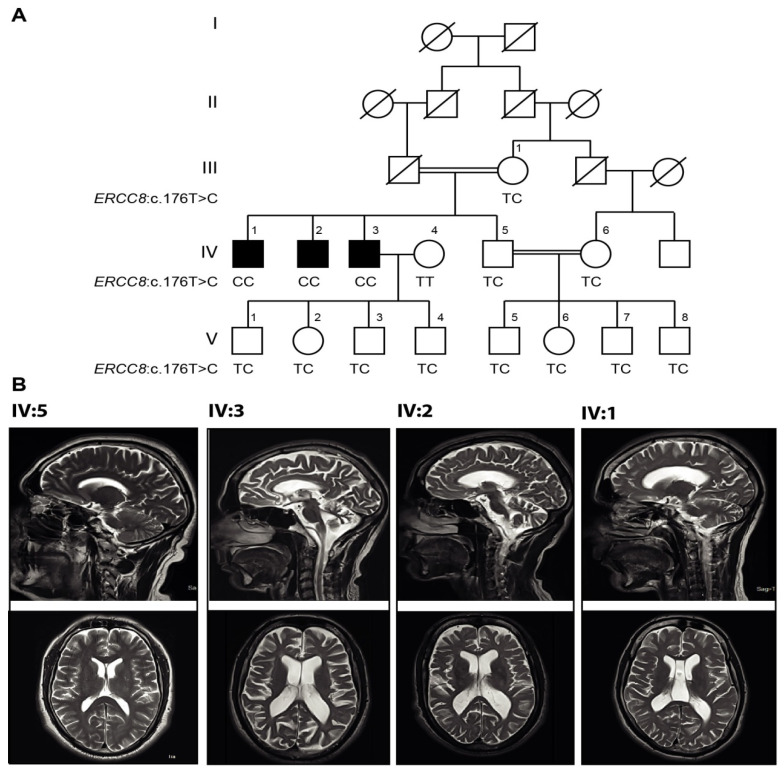
Pedigree, genotypes, and MRIs of the consanguineous family segregating the *ERCC8* variant. (**A**) Pedigree and genotypes of the family segregating *ERCC8* mutation c.176T>C. Squares and circles indicate males and females, respectively. The symbols in black filled represent the affected members. The symbol with a diagonal line indicates a deceased individual. The numbers to the left of pedigree show generation number while those above the symbols denote individuals within that generation. (**B**) T2-weighted brain MRI of the normal brother (IV:5) shows a healthy cerebellum. Patient (IV:3) has prominent cerebellar atrophy/degeneration (upper panel) and prominent intra and extra cerebral CSF spaces (lower panel) shown on axial section of T2-weighted image. Affected brother (IV:2) has prominent cerebellar atrophy/degeneration (upper panel) and mildly dilated intra and extra cerebral CSF spaces (lower panel) shown on axial section of T2-weighted image. Affected brother (IV:1) has very mild atrophy/degeneration of cerebellar folia (upper panel) on axial section of T2-weighted image.

**Figure 2 cells-11-03090-f002:**
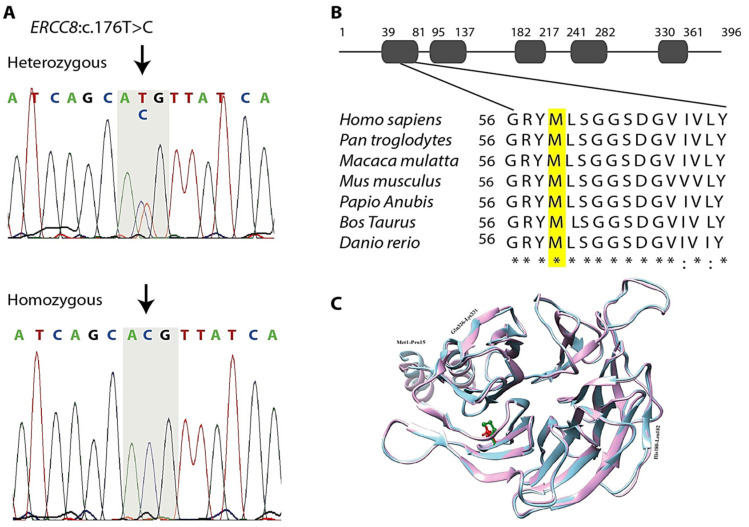
Analysis of the ERCC8^M59T^ variant and its 3D structure modeling. (**A**) Chromatograms of DNA sequence of healthy carrier IV:5 (top) indicates heterozygous for the variant, and affected individual IV:1 (bottom) indicates a homozygous missense mutation in *ERCC8* gene. Arrows indicate the position of the c.176T>C, which resulted in substitution of Methionine (Met, M) to Threonine (Thr, T) at codon 59 (p.M59T). (**B**) Degree of conservation of the Met 59 residue (shaded) across different species. (**C**) Visual comparison of predicted models. ERCC8^WT^ is shown in pink color ribbons, while ERCC8^M59T^ is shown in sky blue color. The regions with conformational changes are labelled in black. Position of M59 is shown green in WT, while red in mutant structure.

**Figure 3 cells-11-03090-f003:**
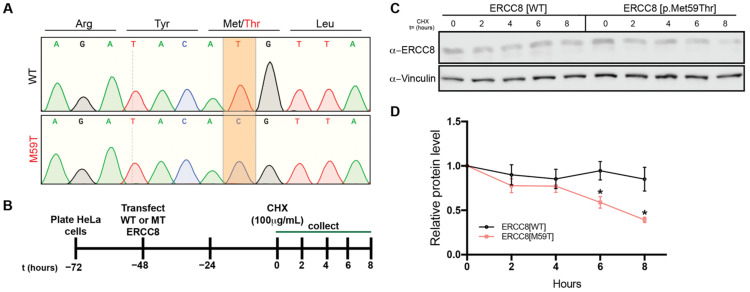
ERCC8^M59T^ mutation decreases protein stability. (**A**) Chromatograms of DNA sequence of plasmids to overexpress full-length ERCC8^WT^ (transcript variant 1) or full-length ERCC8^M59T^. The highlighted nucleotides show the WT sequence (top) and the successfully mutated plasmid (bottom). (**B**) Schematic of ERCC8 protein stability assay. (**C**,**D**) Representative western blots (**C**) and quantification (**D**) of WT and mutant ERCC8 degradation over time (*n* = 3 biological replicates). * *p* < 0.05, *t*-test.

**Table 1 cells-11-03090-t001:** Clinical manifestations of patients with autosomal recessive *ERCC8* mutation.

Patient ID (Sex)	IV:1 (Male)	IV:2 (Male)	IV:3 (Male)
*ERCC8* mutation	c.176T>C homozygous	c.176T>C homozygous	c.176T>C homozygous
Parental Consanguinity	+	+	+
Age of onset (years)	10	12	15
Age at examination	30	37	43
Symptoms at onset	Involuntary movement of hands	Involuntary hands and head movement	Involuntary hands movement
Marital status	Single	Single	Married
Microcephaly	−	−	−
Kyphosis	−	−	−
Ankylosis	−	−	−
Optic atrophy	−	−	−
Growth retardation	−	−	−
Facial features	Normal	Normal	Normal
Sun burn	+	+	+
Cognition	Normal	Normal	Normal
Impaired coordination of legs & arms	+	+	+
Difficulty in writing	+	+	+
Difficulty in handling objects	+	+	+
Dysarthria	+	+	+
Dysphagia	+	+	+
Nystagmus	−	−	−
Diplopia	+	+	+
Visual loss	−	−	−
Grip	Normal	Normal	Normal
Reflexes	Intact	Upper limb depressed, normal knee absent at ankle	Upper limb and ankle jerk depressed, knee jerk positive
Plantar response	Down-going	Withdrawal, nonspecific	Down-going
Finger nose test	+	+	+
Heel shin test	+	+	+
Romberg test	−	−	−
Gait ataxia	+	+	+
Skin pigmentation	Normal	Normal	Hyper
Fallen wound sign	−	−	+
Cranial nerve	Normal	Normal	Normal
Pin prick	Normal	Normal	Normal
others	Divergent squint	Head titubation	None
Cerebellar atrophy	very mild	+	+
Cerebral atrophy	−	+	+

+ Present, − Absent.

**Table 2 cells-11-03090-t002:** List of autosomal recessive homozygous candidate variants derived from exome sequencing data. All variants except one in ERCC8 have been excluded due to no segregation and lack of evidence for brain related function/phenotype.

Gene	Chr	Pos	Ref Allele	Alt Allele	Mutation Type	Aa Change	Allele Frequency	Meta Svm	Cadd Score	%Genic Intolerance Score Based on Exac	Omim/ Diseases	ACMG Classification	Segregation Status
ERCC8	5	60217980	A	G	Missense	p.M59T	0.000008018	D	20.4	53.24	609412/Cockayne syndrome type-A, UV-sensitive syndrome 2	Likely pathogenic (II)	Segregated
MX2	21	42749772	C	A	Missense	p.D102E	0.000007956	D	16.39	29.62	147890/Nil	Variant of Uncertain Significance	No Segregation
PLA2G12A	4	110638815	A	G	Missense	p.Y114H	0.00001989	T	24.9	74.21	611652/Nil	Variant of Uncertain Significance	No Segregation
PLA2G12A	4	110650941	G	A	Missense	p.L9F	0.00002486	T	19.74	74.21	611652/Nil	Variant of Uncertain Significance	No Segregation
GEMIN8	X	14027232	C	T	Missense	p.D177N	Not available	T	16.84	57.84	300962/Nil	Likely benign (I)	No Segregation

## Data Availability

The data presented in this study are available in the main and Appendix A provided in this manuscript. Any additional data are available upon request from the corresponding author.

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
