# Peer review of "A Novel Missense Mutation in *ERCC8* Co-Segregates with Cerebellar Ataxia in a Consanguineous Pakistani Family"

_cells, 2022, doi:10.3390/cells11193090_

Round 1
Reviewer 1 Report
Gauhar et al. wrote a very interesting manuscript entitled " A novel missense mutation in ERCC8 co-segregates with cerebellar ataxia in a cosanguineous Pakistani family". In ocnclusion, the authors stated that theses findings expand the role of ERCC* nutations in autosomal recessive cerebellar ataxias (ARCAs) and indicate that ERCC8-related mutations (Cockayne syndrome and UV light-sensitive syndrome) should be considered in the differential diagnosis of ARCAs. I agree with the authors!! This work sheds more light on the corrrelation between a genotype and its different phenotypes.
Author Response
Please see uploaded file for our response to reviewer's comments

Reviewer 2 Report
The Authors described an interesting case of familial ARCAs, describing a large family with several affected members.
The manuscript is well written and the scientific community needs to report new causative mutations associated with rare diseases.
I would like to point out some suggestions to better clarify specific points:
1) the number of Ethical protocol approved should be reported
2) it would be helpful to report the ACMG (American College of Medical Genomics) classification for the reported variants of interest.
3) Is there an endogenous expression of ERCC8 in HeLa cells? It would be helpful to report it (if there or there isn't) to better clarify the obtained in vitro results
4) I suggest to adding a paragraph with the main limitations of the in vitro study. For example, it is reported a statistically significant reduction of the quantification of WT and Mutant ETCC8 degradation but the le number of replicates is quite low.
Author Response
Please see uploaded file for our response to reviewer's comments.

Reviewer 3 Report
The authors performed a whole exome sequencing in two affected individuals of a consanguineous Pakistani family in order to search for the putative genetic cause of cerebellar ataxia.
Genetic analyses identified a novel homozygous missense mutation in the ERCC8 gene (c176T>C 272, p.M59T), previously linked to autosomal-recessive cerebellar ataxia (ARCA) in one family. The mutation has co-segregated with the cerebellar ataxia phenotype in the family. Multiple in silico tools predicted it was pathogenic. In addition, in silico analysis demonstrated the alteration in the protein's structure of ERCC8 that could affect its associations with other proteins.
The description of Table 2 is puzzling. There are 4 additional genes presented in Table 2 apart from ERCC8. Were they in the homozygous state in the two affected siblings IV:2 and IV:3? It is not stated clearly in the manuscript whether these variants (namely MX2, PLA2G12A, PLA2G12A, and GEMIN8) segregated with the disease phenotype in the family. Could they potentially contribute to the phenotype of cerebellar ataxia or exert a modifying effect?
Experiment with the inhibition of translation (with cycloheximide) in Hela cells overexpressing either WT ERCC8 protein or the one with p.M59T mutation indicated that mutant protein is less stable and has a faster turnover than WT protein. Could you comment on the possible interference of endogenous ERCC8 protein expression with your results?
Altogether this is a well-documented and well-presented case of three patients with cerebellar ataxia whose diagnosis had been based on clinical symptoms and brain MRIs demonstrating cerebellar atrophy. Functional studies (both in silico and in vitro ) speak in favor of the strong contribution of ERCC8 mutation to the described phenotype. The manuscript expands the spectrum of phenotypes related to mutations in the ERCC8 gene.
Author Response

(The authors gave the same response as above.)

Reviewer 4 Report
Title of the paper: “A novel missense mutation in ERCC8 co-segregates with cerebellar ataxia in a consanguineous Pakistani family “
In this study they identified a novel homozygous missense mutation c.176T>C in the ERCC8 gene in a consanguineous ARCA Pakistani family with three affected male siblings. They also reported that the role of ERCC8 33 mutations should be considered in the differential diagnosis of ARCAs.
The authors present a well-written article. The introduction, Materials and Method Figures, Result, Discussion and References are framed correctly. Overall, the article is informative for the scientific fraternity. I recommend the paper for publication with minor correction. The words spacing tin typography to be checked.
Line No. 82: The ethical approval for the study, the sanction reference number should be given
Line No.99: WES (elaboration for the symbol required)
Line No.88: A written informed consent was obtained from the parents. (three affected and eleven unaffected individuals, were recruited. They are young adults. Why the consent form was not taken form the unaffected young adults.)
Line No.111: Public data bases used as
Line No.129: 5 min in a Bio-Rad
Line No. 175: movements of hands and difficulty
Line No. 297: mutation is likely due to by
Line No. 306 & 307: For example, a missense mutation (Ala205Pro) reported in the fourth WD40 motif abolished interaction with DDB1 [28]. (The reference given is not matching)
Author Response

(The authors gave the same response as above.)
